# Position: Uncertainty Quantification Needs Reassessment for Large-language Model Agents

**Michael Kirchhof** [1]  **Gjergji Kasneci** [2]  **Enkelejda Kasneci** [2]

## Abstract

Large-language models (LLMs) and chatbot agents are known to provide wrong outputs at times, and it was recently found that this can never be fully prevented. Hence, uncertainty quantification plays a crucial role, aiming to quantify the level of ambiguity in either one overall number or two numbers for aleatoric and epistemic uncertainty. This position paper argues that this traditional dichotomy of uncertainties is too limited for the open and interactive setup that LLM agents operate in when communicating with a user, and that we need to research avenues that enrich uncertainties in this novel scenario. We review the literature and find that popular definitions of aleatoric and epistemic uncertainties directly contradict each other and lose their meaning in interactive LLM agent settings. Hence, we propose three novel research directions that focus on uncertainties in such human-computer interactions: *Underspecification uncertainties*, for when users do not provide all information or define the exact task at the first go, *interactive learning*, to ask follow-up questions and reduce the uncertainty about the current context, and *output uncertainties*, to utilize the rich language and speech space to express uncertainties as more than mere numbers. We expect that these new ways of dealing with and communicating uncertainties will lead to LLM agent interactions that are more transparent, trustworthy, and intuitive.

## 1. Introduction

Large language model (LLM) agents such as chatbots are notorious for *hallucinating* at times (Bang et al., 2023; Guer-

[1]University of Tübingen. Now at Apple. [2]Technical University of Munich.. Correspondence to: Michael Kirchhof <mail address see website>.

*Proceedings of the 42$^{nd}$ International Conference on Machine Learning*, Vancouver, Canada. PMLR 267, 2025. Copyright 2025 by the author(s).

*Figure 1.* The traditional view on uncertainties suggests a clear black-and-white dichotomy between aleatoric and epistemic uncertainty. We argue that recent developments show this dichotomy is not that simple, and not helpful for developing LLM agents.

reiro et al., 2023), that is, to make up a response that is incorrect. Recent research has shown that this behaviour is rooted in their very generative nature, such that we can not expect LLM agents to stop hallucinating in the future (Banerjee et al., 2024; Kalai & Vempala, 2024; Xu et al., 2024b). Instead, there are numerous approaches to quantify the uncertainty that an LLM agent has in each of its statement, in order to bring transparency to which responses can be trusted and which require further investigation (Kadavath et al., 2022; Kapoor et al., 2024). Such uncertainty quantification methods either output one total uncertainty or, more recently, attempt to output individual values for aleatoric and epistemic uncertainty (Wimmer et al., 2023; Hüllermeier & Waegeman, 2021). *Epistemic uncertainty* is reducible uncertainty, such as when an agent could be trained with more data from new regions of the input manifold to produce more definite outputs. *Aleatoric uncertainty* is irreducible uncertainty, when the data itself is too noisy or lacks features to make predictions that come without a risk of error, regardless of how good the model is. While these uncertainty quantification approaches that assign numerical aleatoric and epistemic uncertainty scalars to each output have been useful more structured tasks such as classification (Murphy, 2012), we argue that they fail to capture the nuanced, multi-turn, and interactive nature of LLM-agent uncertainty in real-world applications.

LLM agents can and must handle uncertainties in a more advanced way. This is because they leave traditional well-

| School of Thought | Main Principle |
| --- | --- |
| Epistemic uncertainty as number of possible models (Wimmer et al., 2023) | Epistemic uncertainty is how many models a learner believes to be fitting for the data. |
| Epistemic uncertainty via disagreement (Houlsby et al., 2011; Gal et al., 2017; Kirsch, 2024) | Epistemic uncertainty is how much the possible models disagree about the outputs. |
| Epistemic uncertainty via density (Mukhoti et al., 2023; Charpentier et al., 2022; Heiss et al., 2022; Liu et al., 2020; Van Amersfoort et al., 2020) | Epistemic uncertainty is high if we are far from seen examples and low within the train dataset. |
| Epistemic uncertainty as leftover uncertainty (Kotelevskii & Panov, 2024; Kotelevskii et al., 2022; Lahlou et al., 2021; Depeweg et al., 2018) | Epistemic uncertainty is the (estimated) overall uncertainty minus the (estimated) aleatoric uncertainty. |
| Aleatoric uncertainty as Bayes-optimal model (Schweighofer et al., 2024; Apostolakis, 1990; Helton, 1997; Bengs et al., 2022) | Aleatoric uncertainty is the risk that the best model *inside a model class* still has, assuming infinite data. |
| Aleatoric uncertainty as pointwise ground-truth variance (Lahlou et al., 2021) | Aleatoric uncertainty is the variance that the output variable has on each input point, and errors because the model class is too simple is *not* part of it. |
| Aleatoric and epistemic as labels of the practitioner (Der Kiureghian & Ditlevsen, 2009; Faber, 2005) | Aleatoric and epistemic are just *terms* with which practitioners communicate which uncertainties they intend to reduce and which not. |

*Table 1.* Overview of prominent schools of thought on aleatoric and epistemic uncertainties and their (conflicting) main principles.

structured setups with fixed-length inputs (one image, one question, or one vector of features) and fixed output formats (one segmentation map (Baumgartner et al., 2019), one answer string (Kwiatkowski et al., 2019b), a single vector of class probabilities (Murphy, 2012)), and are instead applied in much more open environments. In a chat interaction with a user where the user's questions is underspecified and ambiguous, an LLM agent should not only output a numerical uncertainty score, but interact and ask clarification questions. If it detects that it lacks knowledge, it can use retrieval to gather additional information (Lewis et al., 2020), and if there are still remaining uncertainties, it can communicate its uncertainty not only as a number but explain why it is uncertain, which options there are, and what further information can help resolve the uncertainty. This is better suited to the dynamic multi-turn nature of chat interactions where, as we show below, what initially appears as epistemic uncertainty (e.g., lack of knowledge) can become aleatoric uncertainty if additional information fails to reduce the ambiguity, and aleatoric uncertainty (e.g., an underspecified question) can become a reducible epistemic uncertainty by enabling to ask clarifying questions. This motivates our position that **dichotomic views on aleatoric and epistemic uncertainty are inapplicable to modern LLM agent interactions; instead, we need to research how uncertainties in user interactions are detected, handled, and communicated.**

We support this position by contributing three perspectives that aim to inspire creative rethinking of uncertainty quantification in the era of LLM agents:

1. Section 2 provides an in-depth review of the recent developments in aleatoric and epistemic uncertainty disentanglement and *finds that they are fundamentally conflicting*, already in toy-examples. Even if future research could find non-conflicting definitions and decorrelated estimates, we argue that they are not applicable to LLM agent setting, because in a multi-turn exchange between a user and an LLM agent it becomes blurry and ultimately subjective which uncertainties are reducible and which stay irreducible.

2. Section 3 proposes three novel research directions specifically for LLM agent interactions: (1) LLM agent interactions experience strong *underspecification uncertainties*, because not only is much information missing at first, but also even the task itself might be unclear at the start of a conversation, (2) *interactive learning* can help reduce these underspecification uncertainties by interacting with the user, and (3) when it finally comes to communicating the uncertainty, we argue that LLM agents can utilize are more advanced *output uncertainties* than mere answer probabilities.

3. Section 4 takes the counter-position and delineates in which areas traditional epistemic and aleatoric uncertainties and numerical uncertainties remain useful.

We believe that this position, and its counter-position, can help summarize the recent trends in uncertainty quantification and spark a discussion in the larger community.

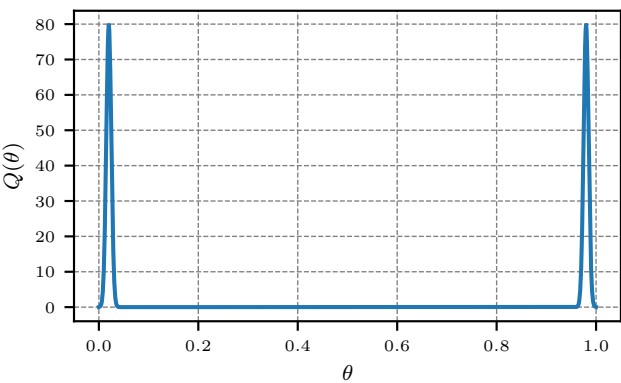

*Figure 2.* In a binary prediction, the learner may have a belief that the Bernoulli probability is either high or low. Some schools of thought see this as a case of maximum epistemic uncertainty whereas other see it as nearly minimal epistemic uncertainty.

## 2. Where Traditional Uncertainties Fail

This section gives an introduction to the traditional research on uncertainties. We critically review the popular dichotomy of aleatoric and epistemic uncertainty and its various realizations, summarized in Table 1. We show that this strict dichotomy has inherent definition conflicts even in simplistic examples (Sections 2.1 to 2.3) and that it breaks apart in modern interactive LLM agent setups (Section 2.4). These arguments support the first part of our position, namely that the traditional terms and methods in uncertainty quantification are unsuitable for LLM agents.

### 2.1. Epistemic Uncertainty: Maximal or Minimal?

Understanding how to quantify epistemic uncertainty is crucial for making reliable predictions, but definitions vary dramatically. This definition conflict that can be seen directly in a simple example. Suppose a learner is parametrized by $\theta$ and models a binary classification problem. In this section, we focus on only one input sample $x \in \mathcal{X}$, so the learner is simply tasked to estimate the probability $p \in [0, 1]$ of a Bernoulli distribution $y|x \sim \text{Ber}(p)$ with the parameter $\theta \in [0, 1]$. We train the learner with some data $\{y_n\}_{n=1}^N$, $y_n \in \mathcal{Y} = \{0, 1\}$, so that it forms a second-order distribution $Q(\theta)$ that tells which parameters it finds plausible for the data. In Bayesian terms, the parameter $\theta$ is a random variable $\Theta$ itself. Suppose that after training, the learner concludes that there are only two possible models left that could fit the data, either $\theta = 0$ or $\theta = 1$, i.e., $Q$ is a mixture of two Diracs, as in Figure 2. Does this reflect a state of maximal or minimal epistemic uncertainty?

There are multiple, equally grounded answers to this question. On the one hand, one can define epistemic uncertainty as a form of *disagreement*. For example, epistemic uncer-

tainty is often defined from a mutual information perspective as $\mathbb{I}_{P(y,\theta|x)}(y; \theta)$ Houlsby et al. (2011); Gal et al. (2017); Kirsch (2024). The mutual information tells how much the variance in $Y$ can be reduced by reducing the variance in $\Theta$. In other words, this epistemic uncertainty formula models how much the possible parameters $\theta \sim \Theta$ disagree in their prediction about $Y$. It follows that the two beliefs $\theta = 0$ and $\theta = 1$ of the learner maximally disagree, and the epistemic uncertainty is maximal.

On the other hand, epistemic uncertainty can be defined based on the number of plausible models that could explain the data. For instance, Wimmer et al. (2023) propose axiomatic definitions of epistemic uncertainty, where the uncertainty decreases as the set of possible models shrinks. Regardless of which specific epistemic uncertainty formula ones derives from them, the axiomatic requirements imply that the epistemic uncertainty must be (close to) zero in our example, because the number of possible models has already been reduced to only two Diracs. In their axiom system, the epistemic uncertainty would be maximal if $Q$ was a uniform distribution. The authors discuss this example in their paper, and, interestingly, there is also a public discussion between the disagreement and the axiomatic parties (Kirsch et al., 2024), which we encourage the curious reader to explore. We also note that being split between $\theta = 0$ and $\theta = 1$ is an extreme example for demonstration purposes, but the example holds for any split belief between two points versus a belief over their convex hull.

Besides these two conflicting schools of thought, there is a third one that relates epistemic uncertainty to how well the training data supports the model's predictions. Under this perspective, epistemic uncertainty does not hinge simply on disagreement or the number of plausible models, but rather on how far we are from well-established data regions. (Mukhoti et al., 2023; Charpentier et al., 2022; Heiss et al., 2022; Liu et al., 2020; Van Amersfoort et al., 2020) define epistemic uncertainty as the *(latent) density* of the training data. This definition has neither a maximal nor minimal uncertainty, since the density values depend on the normalization and prior over the whole space $\mathcal{X}$ (or, analogously, $\mathcal{X} \times \mathcal{Y}$). Hence, in the above example, latent density estimators would answer neither with maximum nor minimum uncertainty but rather 'it depends', namely on how much training data was observed on or close to $x$ in relative comparison to the remaining areas in $\mathcal{X}$, and on the prior that defines how fast and to which value the epistemic uncertainty grows with the distance to the train data.

This shows that epistemic uncertainty is not a universally agreed-upon concept. Different equally well-grounded theoretical foundations lead to contrasting conclusions, even in the above simplistic example, which is both entirely theoretical (leaving estimation errors of the epistemic estimators

aside) and inside one fixed context (the input and output spaces $\mathcal{X}, \mathcal{Y}$ are fixed, and the model class covers all possible data-generating processes). Understanding this diversity of views is essential for navigating real-world scenarios. We will see next that these conflicts do not only occur with epistemic uncertainty.

## 2.2. Aleatoric Uncertainty: Reducible Irreducibility

Building on the definitional conflicts observed in epistemic uncertainty, we now turn to aleatoric uncertainty. Let us expand the above example. We now regard different inputs $x \in [-1, 3]$, and use a linear model that estimates $f(x, \theta) = p(Y = 1 | X = x, \theta)$. Recall that aleatoric uncertainty is often vaguely mentioned as the *irreducible* uncertainty that, even with infinite data, one cannot remove. But what does irreducible mean? There are two major schools of thought: (1) Bayes-optimality proponents who see aleatoric uncertainty as all residual uncertainty within a chosen model class, and (2) data-uncertainty proponents who argue that changing the model class (using more complex functions) can reduce some uncertainties previously deemed aleatoric. This debate is not just philosophical. If a practitioner labels certain uncertainties as aleatoric (and thus, not worth investing in reducing), they may miss opportunities to improve predictions by considering richer model classes or additional data sources.

More precisely, Bayes-optimality proponents formalize aleatoric uncertainty as *the uncertainty that even the Bayes-optimal model has* (Hüllermeier & Waegeman, 2021). However, a Bayes-optimal model is always only optimal within its model class. To quote Schweighofer et al. (2024): *"[t]his [definition of aleatoric uncertainty] assumes that the chosen model class can accurately represent the true predictive distribution"*. In our example, this would be the class of linear models. If the data-generating process was nonlinear, like in Figure 3, this would create leftover risk, called model bias.[1] This is a simple mathematical fact that all theoreticians can agree on, but the question is: Is this irreducible? Bayes-optimality proponents would answer yes; even with infinite data the model bias can not be reduced further, and as irreducible uncertainty, it should be counted towards aleatoric uncertainty. They define aleatoric uncertainty inside the given model class as *"the uncertainty that arises due to predicting with the selected probabilistic model"* (Schweighofer et al., 2024; and similarly Apostolakis, 1990; Helton, 1997). This is also a corollary of axiomatic views that dictate that *"in the limit, i.e., if the sample size goes to infinity, all epistemic uncertainty should disappear"* (Bengs et al., 2022) so that model bias could not be part of the epistemic uncertainty and needs to be counted

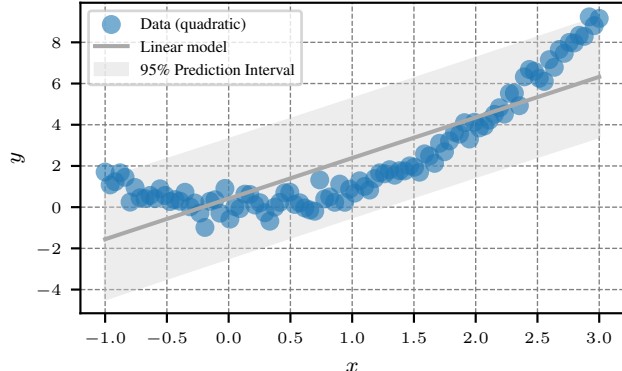

*Figure 3.* Using a too simple model class, like a linear model to fit quadratic data, leads to wide uncertainty estimates. The question is whether this is irreducible, and thus aleatoric uncertainty. Bayes-optimality schools of thoughts would argue that yes, it is irreducible *within the model class* and thus aleatoric, whereas data-uncertainty schools of thought would argue that it is reducible *when choosing a better-suited model class*, hence it is not aleatoric.

towards aleatoric uncertainty. However, as (Hüllermeier & Waegeman, 2021) point out, the choice of a stronger model class may also be considered a means to reduce uncertainty. Hence, the model bias would be a part of the epistemic uncertainty, and aleatoric uncertainty would only be that which no possible model could reduce because the data $X$ lacks the features to make predictions about $Y$. In short, aleatoric uncertainty would be defined as *data-uncertainty* (the *pointwise* Bayes-risk, like in Lahlou et al., 2021), which is *not* the same as irreducible uncertainty (Bayes-optimal within its model class) (Hüllermeier et al., 2022).

Because many frameworks define epistemic uncertainty as whatever remains after accounting for aleatoric uncertainty (Kotelevskii & Panov, 2024; Kotelevskii et al., 2022; Lahlou et al., 2021; Depeweg et al., 2018), the boundary we draw for aleatoric uncertainty (and predictive uncertainty), as well as its estimation, directly shapes our understanding of epistemic uncertainty. Drawing a border on the fuzzy cloud of aleatoric uncertainty directly determines what is considered epistemic uncertainty. This is a consequence of adopting a dichotomous view of uncertainty, where epistemic uncertainty encompasses everything that aleatoric uncertainty does not, without additional categories for factors such as model bias. In short, whether model bias and other forms of reducible uncertainty are classified as aleatoric or epistemic depends on the chosen definitions and model classes, blurring the once-clear line between 'irreducible' and 'reducible' sources of uncertainty.

---

[1]Despite its name, model bias is an uncertainty. It is sometimes referred to as structural uncertainty.

## 2.3. Aleatoric and Epistemic Uncertainty Interplay

Having explored the definitional conflicts in epistemic and aleatoric uncertainties, we now examine what happens when these uncertainties are treated as separate entities. We find that they interplay, which challenges the notion that they can be disentangled.

If aleatoric and epistemic uncertainty were distinct, orthogonal categories (and there were no further categories), one might hope to divide total predictive uncertainty into distinct parts: one reflecting inherent randomness in the data (aleatoric) and another reflecting gaps in our model's knowledge (epistemic). This is proposed by information-theoretical decompositions (Depeweg et al., 2018; Mukhoti et al., 2023; Wimmer et al., 2023), Bregman decompositions (Pfau, 2013; Gupta et al., 2022; Gruber et al., 2023), or logit decompositions (Kendall & Gal, 2017). But does this clean division hold in practice?

For example, Depeweg et al. (2018) define

$$\underbrace{\mathbb{H}_{P(y|x)}(y)}_{\text{predictive}} = \underbrace{\mathbb{E}_{Q(\theta|x)}\left[\mathbb{H}_{P(y|\theta,x)}(y)\right]}_{\text{aleatoric}} + \underbrace{\mathbb{I}_{P(y,\theta|x)}(y;\theta)}_{\text{epistemic}}.$$

(1)

At the first look, the two summands resemble aleatoric uncertainty (average entropy of the prediction) and epistemic uncertainty (disagreement between ensemble members). However, recent empirical studies challenge this clear separation. For example, Mucsányi et al. (2024) demonstrate that, across a wide range of methods from deep ensembles over Gaussian processes to evidential deep learning, aleatoric and epistemic estimators produce values that are almost perfectly correlated, with rank correlations by between 0.8 and 0.999, see Figure 4. This lack of independence means that what we label as 'aleatoric' uncertainty may still serve as a reliable signal for tasks previously thought to be purely 'epistemic,' and vice versa. In practical terms, uncertainty measures intended for one purpose can end up performing well in another domain, blurring the boundaries of their intended roles. Consequently, they observe that the aleatoric uncertainty estimators are about as predictive for out-of-distribution detection (classically considered an epistemic task) as epistemic estimators, and the epistemic uncertainty estimators are as predictive of human annotator noise (an aleatoric task) as aleatoric estimators. Similar observations are made by de Jong et al. (2024) and Bouvier et al. (2022). One may argue that these experimental observations are due to confounded approximation errors and that additive disentanglement is still possible in theory. However, Gruber et al. (2023) assess the formula of a prediction interval of a linear model and denote that *"even in this very simple model one cannot additively decompose the total [predictive] uncertainty into aleatoric and estimation uncertainty"* as the aleatoric (here: observation noise) and epistemic uncertainty

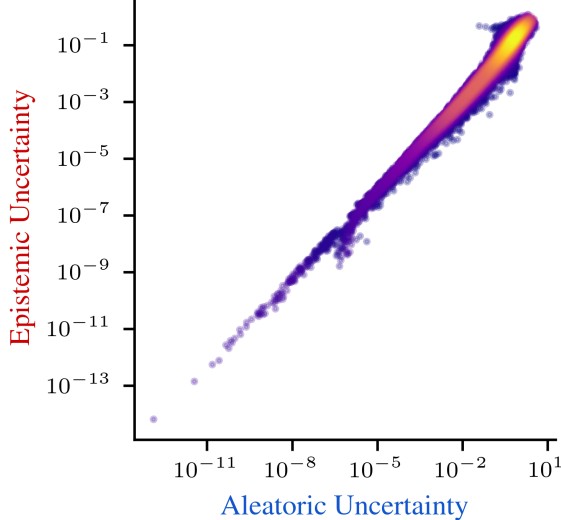

*Figure 4.* When *estimating* aleatoric and epistemic uncertainties, they can often not be disentangled. This plot is reproduced with permission from Mucsányi et al. (2024), where Equation (1) was used to split aleatoric and epistemic uncertainty of a deep ensemble trained on ImageNet-1k. The estimates end up being nearly perfectly correlated, thus capturing the same uncertainty in practice.

(here: approximation error) terms interact non-linearly. The entanglement of the approximation error and the observation noise estimators go further. As Hüllermeier et al. (2022) point out, *"if an agent is epistemically uncertain, it is also uncertain about the (ground-truth) aleatoric uncertainty"*. This is observed in practice by Valdenegro-Toro & Mori (2022) who report that *"aleatoric uncertainty estimation is unreliable in out-of-distribution settings, particularly for regression, with constant aleatoric variances being output by a model. [...] [A]leatoric and epistemic uncertainties interact with each other, which is unexpected and partially violates the definitions of each kind of uncertainty."*.[2]

These practical and theoretical observations lead to the same conclusion, namely, that aleatoric and epistemic uncertainty cannot be split exactly. Most evidence on this is on additive splits, but the latter arguments on epistemic approximation uncertainty about the aleatoric uncertainty estimator (Hüllermeier et al., 2022; Valdenegro-Toro & Mori, 2022) also hold in more generality. To account for these dependencies between aleatoric and epistemic uncertainty estimators, recent methods (Mukhoti et al., 2023) propose to combine multiple estimators. They first gauge if an input point is too far from the training data. They then compute the un-

---

[2]Note that this is not in conflict with Mucsányi et al.'s (2024) findings: Mucsányi et al. find that the aleatoric estimators work well for OOD detection, because on OOD data the aleatoric estimator outputs more flat and thus constantly lower class probabilities, which is similar to what Valdenegro-Toro & Mori (2022) observe in regression.

certainty of the softmax classifier. Each uncertainty has the right to veto and abstain from prediction. This goes to show that often, the actual goal is not to have aleatoric and epistemic uncertainties. Rather, there is a practical task at hand, like abstention, and thinking from this task first and then using different uncertainty estimators, as tools, can solve this task without necessarily labeling one estimator aleatoric and another epistemic.

### 2.4. From Epistemic to Aleatoric and Back: Uncertainties and Chatbots

The concepts of aleatoric and epistemic uncertainty become even more blurred when we go towards agents that interact with the real world. A chatbot is able to ask follow-up questions, which changes the features $x$ responsible for the answer $y$. Let us denote a conversation up to a certain time point $t \in \mathbb{N}$ as some (concatenated) string $x_t$, and let us assume, for simplicity, that the question of the conversation remains the same, so that the true answer distribution $P(Y)$ does not change with $t$. Now that the information that the chatbot gathered in a conversation $x_t$ is dynamic in $t$, is the uncertainty about $Y$ aleatoric or epistemic?

One can argue to only look at fixed time points $t$ in the conversation, where the information $x_t$ collected up to this point poses an irreducible uncertainty for predicting $y$, hence the agent experiences aleatoric uncertainty. Its reduction via follow-up questions would just be a paradoxical illusion as the point $x_t$ in the input space $\mathcal{X}$ for which we calculate the (possibly lower) aleatoric uncertainty changes. However, one can equally argue that – even when still only looking at one fixed point $x_t$ – it is possible to gain more information in future time stepsby further questions or retrieval augmentation (Lewis et al., 2020), so this uncertainty is reducible and epistemic. An argument made by Der Kiureghian & Ditlevsen (2009) (following Faber (2005)), not for chatbots but for sequential modeling in engineering[3], is that the uncertainty may be considered reducible and epistemic until a certain point $t$ when the agent decides to stop asking follow-up questions, which is when it becomes irreducible and aleatoric. That is of course only until the agent finds a new follow-up question to ask and *"the character of the aleatory uncertainty 'transforms' into epistemic uncertainty"* (Der Kiureghian & Ditlevsen, 2009).

Der Kiureghian & Ditlevsen (2009) conclude that calling an uncertainty aleatoric or epistemic is ultimately a subjective choice made by the modeler that just serves to communicate

which uncertainties they attempt to reduce and which not, rather than there being a true aleatoric and epistemic distinction. Similar uncertainties arising from unobserved variables have recently been further studied in the broad sense by Gruber et al. (2023). In the particular sense of natural language processing, these unobserved information paradoxes have lead researchers to propose more general uncertainty frameworks that are *"more informative and faithful than the popular aleatoric/epistemic dichotomy"* because *"[t]he boundary between the two is not always clear cut"* (Baan et al., 2023).

## 3. New research avenues

The previous section shows that estimators for epistemic and aleatoric uncertainty, even when allowing for their ambiguous meanings, can not handle problems that modern LLM agents or chatbots face. In this section, we detail the second part of our position and argue that research is required in three new research directions to handle uncertainties that arise in these novel interaction environments. We group them into three phases of the interaction, namely *underspecification uncertainties* that arise because the input data and the demanded task are not entirely defined by the user, *interactive learning* which allows the chatbot to reduce the underspecification and thus its uncertainties by asking follow-up questions, and lastly communicating *output uncertainties* in its answers that go beyond traditional probability values and utilize the rich expressions that language and speech offer.

### 3.1. Underspecification Uncertainties

In the previous decades of machine learning research, models were defined to solve a specific task, such as classifying a tabular attribute vector with a certain number of features into a finite number of pre-defined categories, or outputting a segmentation map of an image. In any case, the task was fixed and known. Large-language models mark the first multi-purpose tools. They are meant to be generalists, capable of responding to various tasks, which we denote as a finite or infinite set $\mathcal{T}$. The challenge is that it is unknown which task a user has in mind, especially at the start of a conversation, which introduces a first form of *task-underspecification uncertainty*. Mathematically, the distribution over the possible tasks $t$ influences the overall uncertainty over the next token $y$ given the current context $x$ via

$$P(y \mid x) = \int_{t \in \mathcal{T}} P(y \mid t) P(t \mid x) dt. \qquad (2)$$

This shows that the unknown task is a new source of uncertainty, which is, similar to the example in Section 2.4, neither strictly aleatoric nor epistemic. Still, we need to distinguish the task-underspecification uncertainty from other uncertainties in the next-token distribution $P(y|x)$, for ex-

---

[3]We change the example of Der Kiureghian & Ditlevsen (2009) from tabular data to chatbots, because in tabular data adding features changes the input space, so one could argue that it is no surprise that aleatoric and epistemic uncertainty change (Hüllermeier & Waegeman, 2021). In chatbots, the input space is the space of all strings of some finite length and remains the same, and only the input point changes with the timestep.

ample those that arise from lack of knowledge or from semantic equivalences in the token distribution, which all require a different treatment.

A second form of *underspecification uncertainty* is that due to missing input information (Rajpurkar et al., 2018; Zhang et al., 2024b). For example, a user may request *"When did the first Harry Potter movie come out?"*, but without knowing key information such as the country, the answer distribution is highly uncertain. Min et al. (2020) show that ambiguities like this appear in 56% of the test questions in Natural Questions (Kwiatkowski et al., 2019a), a dataset of Google queries. This *context-underspecification uncertainty* is a new variant of the missing variables problems, with the additional edge that there is not just a finite number of columns that could be missing like in a tabular problem, but an infinite number of possible additional context information that could be relevant to the problem. It is up to the LLM agent to select which underspecification uncertainty to tackle, which we describe in the next section. Notably, these context-underspecifaction uncertainies can already arise at a sentence level, when relations between words are vague or sentences can have multiple possible meanings or cultural expectations (Berry & Kamsties, 2004; Kolagar & Zarcone, 2024; Ulmer, 2024).

Current systems are incapable of dealing with these uncertainties, with Zhang et al. (2024c) finding recently that even the best benchmarked model, GPT-3.5-Turbo-16k, can only detect ambiguous questions with 57% accuracy, where 50% is random performance, and human annotators rate only 53% of the follow-up questions as helpful in resolving the ambiguity. This shows that there is a large research gap in how to treat both task- and context-underspecification uncertainties for future interactive LLM agents. Importantly, this problem cannot be "trained away" by relying on a large-enough knowledge base that includes the answer to any question. These uncertainties arise at inference-time, and are due to the user providing insufficient information. This means that even if a future LLM agent was trained on large-enough data to answer any task correctly, it will inevitably face these uncertainties and research is required on how to detect, quantify, and handle them. We go into one research avenue to attempt to handle underspecification uncertainties in the next section.

### 3.2. Interactive Learning

A key characteristic that distinguishes LLM agents from traditional machine learning problems is that LLM agents can interact with the users to learn more about a problem. This could either be to learn information that the LLM has not been trained on (for example, events that happened after its knowledge-cutoff, or information that is private to the user) or information that the LLM knows already, but requires

further information to choose from, because there are underspecification uncertainties. In resemblence to active learning (Settles, 2009), we call this avenue *interactive learning*, where an agent chooses follow-up questions to be able to provide a better answer to the current user interaction.

There are two key characteristics that distinguish interactive learning from active learning: First, the learning is only in order to better solve the current problem $x$, rather than learning about other inputs to improve the overall model $\theta$ as in active learning. Second, in interactive learning, a user is queried for the information rather than a database, which opens research questions in user modeling and human-computer interaction.

First, the agent needs to take into account which information the user can provide. For example, most task-underspecification uncertainties can be resolved by asking the user to clarify their intentions (Zhang et al., 2024a), but missing information that causes context-underspecification uncertainty may also be unknown to the user.

Second, we require human-computer interaction research to judge which follow-up questions to ask and how long to keep asking them (Zhang & Choi, 2023; Pang et al., 2024). As extreme-examples, the LLM may 1) ask too many questions, so that the user loses interest, 2) ask no questions and output a vague or very long answer that covers all possible uncertainties, or 3) impute missing information by the agent's priors and provide an answer implicitly depending on these unknown assumptions. Clearly, neither strategy is optimal. The future question for human-computer interactions thus will be to find the ideal middle ground, that reduces the output uncertainty via follow-up questions without derailing the user interaction. Even if the LLM asked these questions not to a user but to a retrieval system, the ideal trade-off in terms of computational efficiency and latency would remain up for debate.

One side-challenge is that a default LLM agent may ask unnecessary questions, because it has learned in its training data to ask certain questions in certain contexts, although internally it already has the required information. Similar as in active learning, interactive learning approaches may thus incorporate (estimated) mutual-information reductions to choose the best questions to ask, and which answers can already be predicted from the given context with a high-enough certainty.

Interactive learning clearly helps to reduce uncertainties and thus improve the accuracy and personalization (Andukuri et al., 2024) of the overall system. But even besides this obvious advantage, there hides a second important objective, which is accessibility. While a versatile user may provide a clear description of a task with all mandatory side-information to solve a problem, such as a computer scien-

tist providing a clear description of a specified function to add to a given codebase, a less-versatile user, like an elderly person or a young student that uses an LLM agent for education (Kasneci et al., 2023), may require more guidance by the LLM agent. We need not only more research on the optimal strategy to interact with these users, but also more datasets of these user groups, which are under-represented in the current benchmarks that focus mostly on clearly defined one-answer interactions.

### 3.3. Output Uncertainties

Once the LLM agent has determined its underspecification uncertainties, and possibly reduced some of it via interactive learning, it is tasked to communicate all leftover uncertainty in its answer. In its most popular traditional form, this would be a probability shown next to the answer (Lin et al., 2022; Band et al., 2024), or any transformation of it like a binary flag when the LLM is uncertain or a verbalized output that puts the number into pre-defined words (Yona et al., 2024). These questions of calibration and numerical uncertainty scores have attained much attention in early LLM research (Huang et al., 2024; Band et al., 2024) because of their well-founded and well-researched roots in fields like classification. However, we argue that LLM agents have the potential to communicate uncertainties in a much richer way, because they can utilize the whole space of strings rather than the confined space of a single scalar. We label the research that searches for the best ways to communicate leftover uncertainties to a user *output uncertainties*.

A major new opportunity is to not only communicate the overall level of uncertainty, but which competing possibilities there are, why the LLM is uncertain between them, and what could reduce the uncertainty. This can be thought of as an extension of conformal sets and credible intervals (Lee, 1989; Angelopoulos et al., 2023; Kirchhof et al., 2023), with the added challenge of not only outputting a set of answers, but one coherent answer that comprises the possible answers, and with the added opportunity to integrate explainability on why the LLM agent is uncertain between the possibilities (Xu et al., 2024a). Similar to the issue raised in the previous section, it has to be made sure that these possiblities *faithfully* reflect the actual internal belief state of the LLM, and not just common lists of possibilities that the LLM encountered in similar examples in its training data. To ensure this, we encourage to find metrics similar to those in conformal prediction that measure whether the output answer reflects a set of possibilities that is as small as possible but at the same time large enough to cover all likely possibilities. Since the submission of the first draft of this paper, metrics like SelfReflect (Kirchhof et al., 2025) have been proposed for research avenue, along with new methods to generate such summarization strings (Zhang & Zhang, 2025; Yang et al., 2025; Yoon et al., 2025).

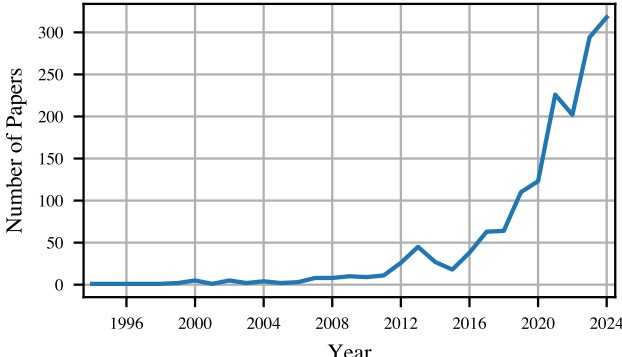

*Figure 5.* ArXiv preprints in computer science, statistics, and math that include the terms "aleatoric" or "epistemic" in their title or abstract. The usage is at an all-time high, with roughly one paper being published each day in 2024.

An LLM agent must also communicate which possibility it finds most likely and which less likely. One avenue to communicate such characteristics is to use verbalized uncertainties such as "most likely", or "perhaps" (Chaudhry et al., 2024; Wang et al., 2024). This is mostly a question of fine-tuning the LLM, but also a question on aligning the meaning of those words to what humans interpret them as, necessitating human-computer interaction research (Van Der Bles et al., 2019; Belem et al., 2024; Steyvers et al., 2024; Belém et al., 2024). When the LLM agent does not communicate with the user via text but via speech, it has even further ways to explicate its uncertainty. The communication of uncertainties via phonetic features (Ulmer, 2024) gives a promising path to utter uncertainties subtly and intuitively to the user.

Overall, LLM agents can benefit substantially from the medium of their outputs, either strings or speech, to better communicate uncertainties. As research on this topic is limited, we encourage the field to find benchmarks and metrics that go beyond scalar uncertainties, so that once these benchmarks are established, the development of uncertainty output methods can be pursued in a quantitative way.

## 4. Alternative Views

Finally, we discuss counter-positions to our arguments and recommendations. We first discuss how aleatoric and epistemic uncertainty are still valuable research avenues (Section 4.1), even for the (pre-)training of LLM agents, how interactive learning could be seen as a normal next-token prediction problem as opposed to a new research avenue (Section 4.2), and in which cases it is beneficial to stick to probabilities to quantify uncertainties, rather than outputting strings that explain the possible answers the agent is uncertain about (Section 4.3).

### 4.1. Aleatoric and Epistemic Uncertainty are Still Valid

One can argue that despite their conflicting definitions, 'aleatoric uncertainty' and 'epistemic uncertainty' still have value as terms and that we should not abandon research on them or using them as labels. We indeed agree with this position: Aleatoric and epistemic uncertainty are terms that are wide-spread, with Figure 5 showing that roughly one preprint is published on arXiv each day that mentions them in the title or abstract. Further, while the quantification of aleatoric and epistemic uncertainty may be less applicable in interactive chat examples (Section 2.4), they still have an importance in the training phase, inside and outside of LLMs, and in particular in choosing which points to query in active learning. We agree that the two terms do allow to easily communicate the rough idea or intention behind an uncertainty quantification approach. Still, when using them, we encourage to follow-up by defining what exactly one intends to use an aleatoric or epistemic uncertainty estimator for and how exactly one defines aleatoric and epistemic uncertainty, to circumvent the naming conflicts in Table 1.

### 4.2. Interactive Learning Can Be Solved By Training on Interaction Data

A counter-position to conducting research on how to learn interactively from a user (Section 3.2) is that interaction behavior could also be learned as a classical next-token prediction task, on interaction datasets (Aliannejadi et al., 2021). For example, one can imagine customer service interactions where the agent demands certain data from the user to fill in a form. Since these interactions are standardized, there should be plenty of training data similar to or even equal to the current context, so that the LLM agent knows which follow-up questions to ask just by reciting past interactions. There would be no need for researching interactive learning.

In this specific scenario, where we have large amounts of interaction training data, we agree that the problem can be mostly addressed by next-token prediction. However, it is still mandatory to prevent the agent from following a trained pattern blindly. For example, a question the LLM is about to ask the user just because it is often asked in this context may have already been answered by the user. The paramount task here remains that the agent's questions need to honestly reflect its internal knowledge. To this date, we lack metrics to capture this, so research on whether the problem is solvable by next-token prediction alone is still required. Further, the human-computer interaction research questions remain: Even if we have datasets of past user interactions, we must ensure that these interactions represent optimal human-computer interaction behavior.

### 4.3. Uncertainties Should be Output as Numbers

In Section 3.3, we make the case that LLM agents need to learn to outline their uncertainties in text rather than in numbers. However, there are also situations in which well-calibrated numbers are preferable. Namely, when the agent is not interacting with a human user but another automated system. For tasks such as abstained prediction, it is simpler to define a threshold value on a numerical predicted uncertainty than to re-interpret what an uttered uncertainty explanation string may indicate. We believe that the two systems, uncertainty as a number and uncertainty as a string, can co-exist, since they are intended for different environments. In a human-computer interaction setup, we expect that an uncertainty that is outlined in text, along with its different possibilities and the reasoning behind them, will provide a better information base to a human decision maker than a mere number, where users may blindly trust outputs when the certainty is high enough. A "blind trust" behavior like this is reported in user testimonies in Kapoor et al. (2024, App. G.3).

## 5. Conclusion

This position paper critically assesses the recent literature in aleatoric and epistemic uncertainty decompositions. Through our examples and references to quantitative and theoretical findings in the literature, we have shown that binarizing uncertainties into either aleatoric or epistemic can create conflicts, and particularly is not supportive for many future applications related to large language model agents. Instead, we expect that research on underspecification uncertainties, interactive learning, and output uncertainties will lead to more transparent, trustworthy, and accessible LLM agents. We encourage the field to take first steps into these directions to build LLM agents that are honest and predictable in their outputs, even when facing complicated contexts with missing data as they are common when interacting with users and the outside world.

## Acknowledgements

The authors would like to thank Kajetan Schweighofer and Bálint Mucsányi. The exchanges on what the true nature of aleatoric and epistemic uncertainty is, if there is any at all, have motivated and shaped this work.

## Impact Statement

This paper presents work whose goal is to advance the field of Machine Learning. We have highlighted potential benefits in transparency and accessibility if our research recommendations are followed in Section 3.2. There are many more potential societal consequences of our work, none which we feel must be specifically highlighted.

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
