# OpenReview forum: "Position: Uncertainty Quantification Needs Reassessment for Large Language Model Agents"
_ICML.cc/2025/Position_Paper_Track — ICML 2025 Position Paper Track poster_

### Official Review · Reviewer_jufP · 2025-03-05

**Significance:** 3
**Argument Clarity:** 2
**Rating:** 2
**Confidence:** 4

**Questions:**

Please see the weakness section.

**Discussion Potential:**

3

**Paper Summary:**

The paper critiques traditional uncertainty quantification methods in machine learning, specifically for large language model (LLM) agents like chatbots. It argues that the standard distinctions between aleatoric and epistemic uncertainty are inadequate for dynamic user interactions.

The authors propose three research directions, including underspecification uncertainties, interactive learning, and output uncertainties.


The aim is to improve transparency and trust in LLM-user interactions.

**Position:**

Yes

**Position In Title:**

Yes

**Related Work:**

3

**Strengths And Weaknesses:**

Strengths:
1. This paper proposes to revisit the uncertainty quantification in the context of LLMs agents, where more advanced requirements exist. The authors present solid evidence and detailed discussion to argue the limitations in the traditional definition (i.e., aleatoric and epistemic uncertainties).
2. The authors provide a new definition and categorization of uncertainty quantification in LLMs agents and provide a detailed discussion.

Weaknesses:
1. I appreciate the three research avenues proposed in this paper, especially the output uncertainty part. But I cannot see a clear motivation of the proposed directions. Is the new categorization stemming from the interaction phase (i.e., before starting the task, trying to solve the task, output decision for the task)? In addition, the authors may need to provide further justification for the new definition of uncertainty quantification in LLMs agents. Overall, I do not get strong evidence that convinces me we should have this new definition/categorization for LLMs agents' uncertainty.
2. Different from the previous definition that is fully supported by the mathematical framework (i.e., aleatoric and epistemic uncertainties), the proposed one seems to be quite empirical. Although it may not be a weakness, the authors should structure the discussion in a more organized manner. The current writing reads more like a discussion in a panel or a presentation instead of a formal definition of a research problem.

**Support:**

2

---

> ### Author Rebuttal · Authors · 2025-04-01
>
> Thank you for your time spent reviewing. We are happy that you find our overview of the conflicting definitions of uncertainty, the first of its kind, helpful. Below, we answer your questions and go into why we structured our position paper like a discussion rather than like a methods paper.
> ### Is the new categorization stemming from the interaction phase (i.e., before starting the task, trying to solve the task, output decision for the task)?
> We make two points in the paper. First, we argue that approaches that try to split aleatoric and epistemic uncertainty into two distinct categories, even when they come with mathematical frameworks, are contradicting each other in theory and in practice (Sections 2.1 to 2.3). We then argue that traditional treatments of uncertainty, already problematic in simple synthetic tasks, are particularly ill-suited to LLM chatbot settings, where the challenges become more pronounced (Sections 2.4 to 3.3).
>
> We thus propose to look at LLM chatbot uncertainties from three perspectives, underspecification, interactive learning, and output uncertainties. This is indeed motivated from the flow of any chat interaction: Underspecification is concerned with user inputs only, interactive learning with the interaction between users and chatbots, and output uncertainties are concerned only with how the chatbot drafts its response to communicate the uncertainty. We refrain from calling this a categorization in the sense of “mutually exclusive” (which we critique in our position for aleatoric and epistemic uncertainty categorizations), but we indeed believe that these rough groups will aid the future research papers and remain up-to-date, since they follow the problem of chatbot interaction itself rather than a specific approach, which might become outdated or context-dependent.
>
> ### Different from the previous definition that is fully supported by the mathematical framework (i.e., aleatoric and epistemic uncertainties), the proposed one seems to be quite empirical.
> We want to highlight that we intentionally do not propose a single definition, since we submit this paper as a position paper to spark discussions in the community, rather than a methods paper to define a new framework. The purpose of our position paper is to give the first literature overview of how uncertainty definitions, even if they are internally mathematically defined, contradict each other, even approaches that are equally well-defined. We do this in sections 2.1 to 2.4 both by running simple mathematical counterexamples through the existing frameworks, and by connecting empirical insights from recent large-scale investigations. We believe that this combination of theoretical and empirical arguments is a strong foundation for a position paper.
> ### The current writing reads more like a discussion in a panel or a presentation instead of a formal definition of a research problem.
> As outlined above, this is intended, following the call for papers of the position paper track. The goal of our position paper is to give the broader ICML audience an unbiased insight into the recent findings of expert papers on uncertainty quantification. This is why we quote all different “schools” of uncertainty estimation in Sections 2.1 to 2.4 (and Table 1) neutrally, and give multiple “Alternative Views” in Section 4, which is required as per the call for papers of the position paper track.
>
> **Thank you again for your time** during the busy rebuttal period. We hope that the above helped underline that, as a position and not a methods paper, our goal is to bring the recent discussions within the uncertainty quantification community to the broader ICML audience. If you believe in the utility of this overview for the position paper track, we would be happy if you reconsidered your score.

---

### Official Review · Reviewer_Sgod · 2025-03-06

**Significance:** 3
**Argument Clarity:** 3
**Rating:** 4
**Confidence:** 3

**Questions:**

I don't have any question.

**Discussion Potential:**

3

**Paper Summary:**

This position paper argues that the traditional dichotomy of aleatoric uncertainty (irreducible randomness in data) and epistemic uncertainty (reducible uncertainty from limited knowledge) is inadequate for large language model (LLM) agents operating in interactive settings. The authors demonstrate that existing definitions of these uncertainty types are fundamentally contradictory even in simple examples and become especially problematic in multi-turn conversations where uncertainty boundaries blur. They propose three new research directions to address uncertainties in LLM agents: (1) underspecification uncertainties for when users provide incomplete information or unclear tasks, (2) interactive learning for asking follow-up questions to reduce uncertainty, and (3) output uncertainties to better express nuanced uncertainty through language rather than just numerical scores. These approaches would better reflect the dynamic, context-dependent nature of uncertainty in human-AI interactions and lead to more transparent, trustworthy, and intuitive LLM agent behavior.

## update after rebuttal
I find the paper interesting and timely, and I appreciate its attempt to reframe uncertainty quantification in LLMs. However, I agree with reviewer Y4Ty that the current scope is narrow and more focused on conversational settings. I will maintain my positive score but do not strongly champion the paper.

**Position:**

Yes

**Position In Title:**

Yes

**Related Work:**

4

**Strengths And Weaknesses:**

Strengths

Clear position articulation: The paper clearly states its position that "traditional views on aleatoric and epistemic uncertainty are inapplicable to modern LLM agent interactions" and proposes a new framework focusing on underspecification uncertainties, interactive learning, and output uncertainties.

Thorough literature review: The paper provides a comprehensive review of existing uncertainty quantification approaches, as evidenced by Table 1 which maps different schools of thought and principles. This demonstrates the authors' deep understanding of the field.

Compelling examples: The authors provide concrete examples demonstrating the definitional conflicts in uncertainty quantification, such as the simple binary classification example that shows contradicting interpretations of epistemic uncertainty (Section 2.1).

Weaknesses

Implementation Challenges Not Fully Addressed: The position paper offers innovative research directions for uncertainty quantification in LLMs but falls short in addressing the practical challenges that may arise during implementation. For instance, it remains unclear how such uncertainties, when expressed in varied and potentially subjective natural language forms, would be standardized across different systems or interpreted consistently by different models. Additionally, the paper does not explore the computational implications of these methods, such as the potential increase in processing time or the complexity of training models to understand and generate these nuanced outputs. Addressing these challenges is crucial for advancing the practical application of the proposed uncertainty quantification methods and ensuring they can be effectively integrated into operational LLM systems, thereby enhancing their reliability and user trust.

**Support:**

3

---

> ### Author Rebuttal · Authors · 2025-04-01
>
> Thank you for your review and for praising our compelling examples we give in our thorough and first literature overview of our position that the popular aleatoric and epistemic uncertainties used by many practitioners fall short in more recent theoretical and practical investigations. We aim to bring the most recent findings from the expert uncertainty quantification community to a broader audience via the position paper track. We address the points you have raised below.
> ### Practical challenges that may arise during implementation
> We intentionally do not want to make a specific mathematical proposition as to how uncertainties should be quantified or communicated, see also Section 3.3, since we would see this as a part of a methods paper rather than a position paper. Our position is that uncertainties are highly context dependent, so similarly to how aleatoric and epistemic uncertainty are no fixed terms, our position, defining underspecification uncertainties would require defining the exact usage context. Our goal with this position paper is to inform about such subtleties rather than provide concrete methods, which in our view could and should be done in methods papers.
>
> ### Computational implications
> Thank you for bringing up this important point. The submitted paper starts a discussion on this in the fourth paragraph of Section 3.2, with an explicit focus on the cost trade-off for the user, since they will probably be the most limiting time factor (and since a chatbot needs to prevent user exasperation). We will include the discussion of the computational trade-off on the LLM side into this paragraph in the camera-ready version.
>
> **Thank you again for your enthusiastic review.** We hope that your acceptance score allows us to open the discussion of aleatoric and epistemic uncertainties in the context of LLM chatbot agents to the broader ICML audience.

---

### Official Review · Reviewer_Y4Ty · 2025-03-15

**Significance:** 3
**Argument Clarity:** 3
**Rating:** 2
**Confidence:** 4

**Questions:**

This position paper is very timely and addresses important research directions for uncertainty quantification in LLM applications. From a practical viewpoint, applying the concept of data and model uncertainty in everyday use cases is difficult. Say we observed multiple samples from LLMs while solving a QA problem. How can we quantify uncertainty?

If we walk away from a very well-defined theoretical research setting and if we see papers published in recent application-oriented conferences, each research paper defines uncertainty quantification in its way per use case. Sometimes, it is confidence, which is not well-defined, entropy measured in a certain way, probability estimate of correct outcome, calibration error, etc. The lack of a unified view of "what we are solving" is a big problem. This paper addresses this issue, and I agree with this view. It is implied from many existing researches. For example, the underspecification uncertainty is closely related to selective generation. The interactive learning is well connected to RAG. The output uncertainty is also well connected to judge or the conformal prediction research.

Let's say that we want to re-assess UQ for LLM agents.
What will be the scope of such consideration?
Is this mainly applicable to a conversational setting? Do we apply it to comprehension tasks? Do we use it for long-form generation tasks?

Many recent papers consider hallucination, which is also a term that is not well defined. Could you make some connections to the research directions for hallucination and uncertainty quantification?

For each research direction, how can we measure the degree of uncertainty? What are the criteria that we can use to improve or reduce uncertainty?

As mentioned above, the literature addresses those research topics, at least partially, under different categorizations. Should we re-group such problems under the three proposed directions? Could you further justify this position?

**Discussion Potential:**

4

**Paper Summary:**

The position of this paper is to propose a new viewpoint on uncertainty quantification around the large language model research.
Section 2 reviews the conventional concepts about epistemic and aleatoric uncertainty and raises the question of whether it can be well divided, as the theory suggests.
Section 2.4 illustrates some examples of why this separation is not suitable for studying uncertainty quantification around large language model applications.
Section 3 proposes three divisions on research areas for uncertainty quantification. The first is related to the input to LLMs, the second is related to the interaction with LLMs, and the last is related to the final answer from LLMs.
Lastly, this paper also reviews the counter position that the conventional viewpoint may still be valid based on the number of publications mentioning the terminology and particular scenarios.

**after rebuttal*
The topic and the position are interesting.
However, the coverage of this paper is narrow.
There are many existing UQ in LLM that could be considered in this position paper, but it is limited to a conversational setting.
During the rebuttal, the authors elaborated in more detail on how we can measure the degree of uncertainty in a new proposed setting. However, it is applying existing techniques for each stage.
I think both the coverage of the literature and applicable UQ problems need to be broadened, and it should suggest or compare the method or metrics of UQ that need to be considered more.
I will keep the current score.

**Position:**

Yes

**Position In Title:**

Yes

**Related Work:**

1

**Strengths And Weaknesses:**

The strength of this paper is as follows.
(1) This paper summarizes existing concepts in uncertainty in Table 1. Section 2 explains those schools of thought with examples and backs up the position of this paper.
(2) This paper also gives an intuitive example using a chatbot to motivate why the proposed position is proper.

The weakness of this paper is as follows.
(1) This paper doesn't show many existing uncertainty quantification research for LLMs. This paper's only connection is in Section 2.4 through an example.
(2) Let's imagine we accept this position and do research as suggested in this position paper. The three cases that are enumerated in this paper are more like topical issues.

**Support:**

3

---

> ### Author Rebuttal · Authors · 2025-04-01
>
> Thank you for your reviewing time and for recognizing our discussion as very timely. We are happy to respond to the questions you have raised below.
> ### From a practical viewpoint, applying the concept of data and model uncertainty in everyday use cases is difficult. Say we observed multiple samples from LLMs while solving a QA problem. How can we quantify uncertainty?
> There are basically two approaches to this in the current literature: Numerical uncertainties in the form of a tuple (greedy-decoded answer, percentage) and verbalized uncertainties that turn the percentage in the tuple into a string (“I am not very sure but…”). We believe that the focus on percentages comes from UQ’s roots in classification, where traditional calibration rightfully plays an important role. However, in this position paper we want to discuss the point that such estimates are not enough anymore when an LLM chatbot agent communicates with a user. To summarize our points in Sections 3.1 to 3.3, we believe that once an agent detects uncertainty, e.g. by sampling from itself multiple times, e.g., via Chain-of-Thought reasoning, it should ask questions for information that it believes the user can give, and once it has gathered all data, outline which possibilities remain, and which further data could resolve them. It is not our goal in this paper, however, to give explicit methods for this, since we want to distinguish our position paper from methods papers.
>
> ### Let's say that we want to re-assess UQ for LLM agents. What will be the scope of such consideration (conversation / comprehension / long-form generation)?
> These are good points. We indeed believe that UQ reconsiderations for LLM agents should primarily focus on chat setting. This is because the other two points follow from it: 1) Comprehension is intrinsic due to context-dependence, and 2) Chats are intrinsically long-form generation (with potential context shifts), compared to the one-token or one-sentence answers in question answering.
>
> We would like to note, however, that our paper also gives the first literature overview of the issue of contrasting aleatoric and epistemic uncertainties in the general setting. We have carefully chosen the most simplistic setups possible in Sections 2.1 to 2.3 to show that the chat setting is a special case of more general problems, so that improvements there will also trickle down to other fields.
> ### For each research direction, how can we measure the degree of uncertainty? What are the criteria that we can use to improve or reduce uncertainty?
> Let us answer for each research direction individually, namely 1) Detecting underspecification, 2) Interactive Learning, and 3) Output uncertainties. When it comes to detecting underspecification, there are two challenges: i) Quantifying that there is uncertainty, which can be handled, e.g., by sampling multiple times, and ii) Deciding whether a user could provide information that would resolve them. Here, the best approach available to an LLM chatbot agent is probably to simulate the chat ahead and re-estimate if there is uncertainty left. This also already goes into 2) interactive learning. A hard objective to quantify is whether an interaction promises to reduce uncertainty would for example be estimating the pointwise mutual information (i.e., finding the Bayes-optimal intervention). For 3), output uncertainties, the objective is less to reduce the uncertainty but rather to communicate it clearly. As we note in the paper, this is more a question for human-computer interaction research.
>
>
> ### As mentioned above, the literature addresses those research topics, at least partially, under different categorizations. Should we re-group such problems under the three proposed directions? Could you further justify this position?
> We believe our research avenues of underspecification, interaction, and output is fitting since it roughly follows the user-input $\rightarrow$ user-chatbot-interaction $\rightarrow$ chatbot-output pattern of chat interactions. It is thus not bound to a specific approach but to the problem itself and should remain in generality. However, as we note in our position, a main argument of this paper is to shed light on the issue that uncertainties are not the same in each context. Any overly strict adherence to a categorization, whether into aleatoric and epistemic or into our three research avenues, will ultimately become mis-aligned because of this. Thus, in alignment with our position, we argue that our research avenues may serve well as rough terms to quickly communicate the purpose of a paper, but need to be unremittingly followed up with a more concrete definition in the context of the individual paper.
>
> **Thank you once again for the discussion.** If you believe that the discussion and the above points are useful to present to the general ICML audience, we would be happy if you considered changing your score to allow for an acceptance.

---

### Official Review · Reviewer_Ph9r · 2025-03-15

**Significance:** 3
**Argument Clarity:** 3
**Rating:** 3
**Confidence:** 3

**Questions:**

How should underspecification uncertainty be quantified? What are the trade-offs between interactive learning and efficiency? How do different user types perceive uncertainty explanations? Could underspecification uncertainty be addressed through prompt engineering?

**Discussion Potential:**

2

**Paper Summary:**

This position paper evaluates the traditional framework of uncertainty quantification, particularly the distinction between aleatoric and epistemic uncertainty, in the context of LLM agents. The authors argue that this classical dichotomy is insufficient for the open-ended, interactive nature of LLM-based interactions. They identify conflicts in existing definitions and propose three new research directions for uncertainty quantification in LLM agents.

**Position:**

Yes

**Position In Title:**

Yes

**Related Work:**

2

**Strengths And Weaknesses:**

Strength:

1. It is a well-motivated new research directions.

2. The paper provides interdisciplinary Insights.

Weakness:
1. the author introduces the term: "Underspecification Uncertainty", but how it should be quantified and measured remains unclear.

2. The paper proposes that LLMs should actively ask clarifying questions, but does not discuss the computational trade-offs.

3. In the alternative view section. While the paper briefly acknowledges scenarios where aleatoric/epistemic uncertainty remains useful, a deeper discussion on why these concepts have persisted despite their flaws would strengthen the argument.

4. The paper claims that rich textual explanations of uncertainty are preferable, but how users perceive and interpret such explanations is not explored in detail. How should explanations be formatted for different user expertise levels (e.g., expert users vs. laypeople)?

**Support:**

2

---

> ### Author Rebuttal · Authors · 2025-04-01
>
> Thank you for your time spent reviewing and for recognizing our argument that a clear distinction between aleatoric and epistemic uncertainty is not well-suited for LLM chat scenarios. We are happy to jump into the discussion below, which we see as the most important part of any position paper.
>
> ### The author introduces the term: "Underspecification Uncertainty", but how it should be quantified and measured remains unclear.
> We intentionally do not want to make a specific mathematical proposition as to how underspecification should be specified, since we would see this as a part of a methods paper rather than a position paper. Our position is that uncertainties are highly context dependent, so similarly to how aleatoric and epistemic uncertainty are no fixed terms, our position, defining underspecification uncertainties would require defining the exact usage context. Our goal with this position paper is to bring up the discussion that future research should focus on important definitions like this rather than us proposing one definite one.
>
>
> ### The paper proposes that LLMs should actively ask clarifying questions, but does not discuss the computational trade-offs.
> Thank you for bringing up this important point. The submitted paper starts a discussion on this in the fourth paragraph of Section 3.2, with an explicit focus on the cost trade-off for the user, since they will probably be the most limiting time factor (and since a chatbot needs to prevent user exasperation). We will include the discussion of the computational trade-off on the LLM side into this paragraph in the camera-ready version.
>
> ### How should explanations be formatted for different user expertise levels (e.g., expert users vs. laypeople)?
> We raise this question in the last paragraph of Section 3.2. Ultimately, it is not only a question of format but also a question of the prior knowledge that (an LLM predicts that) a user has. There are first approaches on this from personalization literature, which we cite, but it is not our goal with this position paper to propose approaches as in a main track paper, rather to open the discussion and outline research avenues that are underexplored.
>
> ### While the paper briefly acknowledges scenarios where aleatoric/epistemic uncertainty remains useful, a deeper discussion on why these concepts have persisted despite their flaws would strengthen the argument.
> As we show in Figure 1, the terms indeed enjoy popularity. We believe, as we note in Section 4.1, that this is due to their apparent clear definition, which makes them attractive especially to outsiders of the uncertainty field, and because they have been proposed early enough in deep learning literature (in 2017, [1]), to spread. However, there are more recent findings coming from inside the uncertainty estimation field that show that these definitions are mal-defined, see [2, 3, 4, 5]. It is our goal with this position paper to open these technical findings from inside the expert community to the broader scientific literature and raise the awareness about this issue, which we are, to our knowledge, the first to give a literature overview over.
>
> **Thank you once again for the discussion.** We hope that the above discussions clarify how our position paper is intended to bring the most recent findings in the expert community of uncertainty estimation to a broader audience. If you agree on the importance of this discussion, we would be happy if you considered revising the score.
>
> ---
>
> [1] Yarin Gal, Riashat Islam, and Zoubin Ghahramani. Deep Bayesian active learning with image data. International Conference on Machine Learning, pp. 1183–1192. PMLR, 2017.
>
> [2] Bálint Mucsányi, Michael Kirchhof, and Seong Joon Oh. Benchmarking uncertainty disentanglement: Specialized uncertainties for specialized tasks. Advances in Neural Information Processing Systems (NeurIPS), 2024.
>
> [3] Ivo Pascal de Jong, Andreea Ioana Sburlea, and Matias Valdenegro-Toro. How disentangled are your classification uncertainties? arXiv preprint arXiv:2408.12175, 2024.
>
> [4] Victor Bouvier, Simona Maggio, Alexandre Abraham, and Léo Dreyfus-Schmidt. Towards clear expectations for uncertainty estimation. arXiv preprint arXiv:2207.13341, 2022.
>
> [5] Cornelia Gruber, Patrick Oliver Schenk, Malte Schierholz, Frauke Kreuter, and Göran Kauermann. Sources of uncertainty in machine learning–a statisticians’ view. arXiv preprint arXiv:2305.16703, 2023.

---

### Decision · Program_Chairs · 2025-05-01

**Decision:**

Accept (poster)

**Comment:**

As one reviewer says:  This position paper is very timely and addresses important research directions for uncertainty quantification in LLM applications.  Reviewers make suggestions for improvement.
One reviewer mentions more connections with hallucination research be included.  One reviewer suggests the paper should be broadened, and doesn't include enough other uncertainty quantification research for LLMs.  Valid criticisms.
Both Weak Reject reviews are confident in their assessment.
The categorisation of three directions is useful but one reviewer argues that they lack clear motiviations.
The paper, however, retains great discussion potential.